# Total Polyunsaturated Fatty Acid Level in Abdominal Adipose Tissue as an Independent Predictor of Recurrence-Free Survival in Women with Ovarian Cancer

**DOI:** 10.3390/ijms24021768

**Published:** 2023-01-16

**Authors:** Helene Salaun, Mathilde Poisson, Adeline Dolly, Flavie Arbion, Stéphane Servais, Jean François Dumas, Caroline Goupille, Lobna Ouldamer

**Affiliations:** 1Department of Oncology, Centre Hospitalier Régional Universitaire de Tours, Hôpital Bretonneau, 2 Boulevard Tonnellé, 37044 Tours, France; 2Laboratoire «Nutrition, Croissance et Cancer», INSERM UMR1069, Université de Tours, 10 Boulevard Tonnellé, 37044 Tours, France; 3Department of Pathology, Centre Hospitalier Régional Universitaire de Tours, Hôpital Bretonneau, 2 Boulevard Tonnellé, 37044 Tours, France; 4Department of Gynecology, Centre Hospitalier Universitaire de Tours, Hôpital Bretonneau, 2 Boulevard Tonnellé, 37044 Tours, France

**Keywords:** adipose tissue, ovarian cancer, polyunsaturated fatty acids, recurrence-free survival, fatty acid metabolism

## Abstract

Prognostic factors for epithelial ovarian cancers (EOCs) are in particular clinical factors such as pathology staging at diagnosis (FIGO stages), genetic mutation, or histological phenotypes. In the present study, FIGO stage, tumor residue after surgery, and body mass index were clinical predictors of recurrence-free survival (RFS). Nonetheless, a number of studies support a lipid metabolism disorder in ovarian cancer patients. The objective of this pilot study was to explore whether fatty acid composition of adipose reflecting the qualitative dietary intake and fatty acids metabolism may be associated with RFS. Forty-six women with EOCs and six with borderline ovarian tumors between March 2017 and January 2020 were included in this prospective study at Tours university teaching hospital (central France). The patients involved in the present study are part of the METERMUS trial (clinicaltrials.gov NCT03027479). Adipose tissue specimens from four abdominal locations (superficial and deep subcutaneous, visceral (pericolic), and omental) were collected during surgery or exploratory laparoscopy. A fatty acid profile of adipose tissue triglycerides was established by gas chromatography. Fatty acids composition was compared among the four locations using nonparametric Friedman’s ANOVA test for repeated measures. Median follow-up of EOC patients was 15 months and patients’ RFS was analyzed using Kaplan–Meier survival curves and log-rank test by separating patients into two groups according to median fatty acid levels. The content of long-chain saturated fatty acids (SFAs) was increased and that of long-chain polyunsaturated fatty acids (PUFAs) decreased in deep versus superficial subcutaneous adipose tissue in EOC patients. Nevertheless, the content of total SFAs was ~28%, monounsaturated fatty acids (MUFAs) ~55%, PUFAs n-6 ~11.5%, and PUFAs n-3 about 1.3%, whatever the adipose tissue. When EOC patients were separated into two groups by median fatty acid content, total PUFAs (n-6+n-3) levels, whatever the adipose tissue, were positively and independently associated with RFS. RFS was about two times longer in EOC patients with high versus low total PUFA content (median survival: 12 vs. 27 months, *p* = 0.01 to <0.0001 according to the tissue). Content of total PUFAs (n-6+n-3) in abdominal adipose tissue (visceral and subcutaneous) are new prognostic factors in EOC.

## 1. Introduction

With 295,414 new cases per year worldwide, ovarian cancer (OC) is the seventh most common cancer in the female population. The age-standardized incidence rates are greater in high-income counties such as those in Europe and North America (8–11/100,000 women) and lowest in Asia and Africa (about 5/100,000 women) [1,2,3].

During 40 years, almost 90% of women have presented malignant OC with an epithelial origin (epithelial ovarian cancer (EOC)); the median age at diagnosis is about 65 years. Although classified as EOC, a high proportion of EOC cases could originate from the fallopian tube epithelium [3]. Indeed, epidemiological studies have described decreased risk of OC with tubal ligation or hysterectomy [1]. OC is often called a “silent killer” because most patients do not exhibit disease-specific symptoms until an advanced disease stage (dissemination to peritoneal cavity), which thus increases the risk of metastatic spread and early death [4].

EOC comprises several histological subtypes with different risk factors, genetic background, clinical course, sensitivity to chemotherapy, and prognosis. EOC is mostly represented by the high-grade serous carcinoma (HGSC) histological subtype, with 70% incidence; these tumors develop de novo from tubal and/or ovarian surface epithelium [5] and present strong chromosomal instability [6]; 40% of cases present hereditary BRCA1 and BRCA2 mutations. Endometrioid carcinoma (about 10% incidence) and clear-cell carcinoma (about 10% incidence) are believed to arise from endometriosis. Low-grade serous carcinomas (with <5% incidence) are genetically stable and characterized by a low number of genetic mutations; they develop slowly from precursors and behave in an indolent manner. Mucinous carcinoma with KRAS mutations are reported in cystadenomas, borderline tumors, and mucinous carcinomas [7].

The current available treatments are surgery and cytoreduction followed by platinum/taxane-based chemotherapy. Bevacizumab added to platinum-based chemotherapy showed an overall survival benefit in patients with poor prognosis [8,9]. More recently, maintenance therapy with PARP inhibitors (olaparib) showed substantial benefit for progression-free survival among women with newly diagnosed advanced OC and BRCA1/2 mutation, with a 70% lower risk of disease progression or death with olaparib than placebo [10]. However, the response to chemotherapy is heterogeneous according to histological subtype (good response to cytotoxic agents for HGSC, poor response for clear-cell and low-grade serous carcinoma). HGSC remains associated with very poor prognosis [3,7]. About 75% of patients with advanced disease will eventually show relapse [7,11]

The factors that may define the risks of recurrence or survival remain to be refined. Age (<65 years), low-performance status, early FIGO stage, some histotypes, or absence of tumor residue after surgery are good prognostic factors [11]. Body mass index (BMI) impact is still unclear, but unfavorable prognosis is observed when the weight is lower than normal or the muscle mass index is low at the time of diagnosis [12,13]. The recurrence risk is increased if the initial OC is advanced at the time of diagnosis, which is why OC is sometimes referred to as a chronic or incurable disease.

Observations that women with the same cancer characteristics given similar treatments can have different outcomes suggest other factors in addition to unmodifiable cancer characteristics that can affect survival. Some studies report that diet quality could be associated with recurrence or mortality [14]. Two studies of OC concluded that high diet quality with low fat and fruit/vegetables consumption (not a Western diet) leads to reduced mortality [15,16]. In this last Playdon study [16], nutrient analysis of lipid quality details found high content of saturated fatty acids (SFAs) associated but not significantly with increased mortality (*p* = 0.07) and high polyunsaturated/monounsaturated fatty acid (PUFA/MUFA) ratio with increased survival (*p* = 0.06). However, diet was estimated with a food-frequency questionnaire.

Several recent studies indicate an alteration of lipid metabolism in ovarian cancer cells, with an increase in endogenous lipid synthesis and fatty acid absorption by tumor cells to meet their high energy demand [17,18]. The relationship between ovarian cancer cells and visceral adipose tissue is a typical feature of ovarian cancer, which induces metabolic adaptation of the tumor to a lipid-rich environment and promotes ascites formation and metastasis. Reducing fatty acid import by neutralizing CD36 or FABP4 transporters in tumor cells controls tumor progression and metastasis, and increases sensitivity to chemotherapy [19,20]. Future therapeutic strategies may target lipid metabolism [21,22], but exploration of fatty acid metabolism remains uncommon in ovarian cancer patients. Reduction in plasma LDL and polyunsaturated glycerophospholipids was noted in ovarian cancer patients with shorter survival [23,24]. We noticed only one article that reported the consequences of ovarian cancer on adipose tissue composition with a mobilization of linoleic acid from subcutaneous and omental adipose tissues [25]. The fatty acid composition of adipose-tissue fatty acids is considered to reflect dietary habits. A number of studies proposed adipose composition as a relevant biomarker to assess past lipid intake quality. Indeed, it reflects lipid intake during several months [26] because of the slow turnover of fatty acids; the half-life has been estimated between 1 and 2 years in healthy individuals with stable weight [27,28]. However, this turnover is dynamic in order to adapt to pathophysiological changes in the body, starting with weight loss or gain. Thus, the greater the lipolysis, the faster the turnover, and vice versa [28,29].

We hypothesized that fatty acid composition of adipose tissue may be impacted by the strong lipid uptake and fatty acid oxidation of ovarian cancer cells, and that adipose fatty acid profiles may be associated with outcomes in women with EOC. We studied the fatty acid profiles of samples from four different abdominal adipose tissues in a prospective cohort and investigated whether the composition differed according to EOC recurrence.

## 2. Results

In this study, 52 patients were included: 46 with EOC and 6 with borderline ovarian tumors (Table 1). Median age was 67.5 years and median BMI was 26.6 kg/m^2^, which highlights that more than 50% of the population was overweight or obese. The EOC and borderline tumor populations did not differ in age, BMI, menopausal status, hormonal context (parity or menopause status), or pathologies commonly associated with overweight such as high blood pressure, diabetes, or dyslipidaemia. Serous carcinoma was the major histological phenotype (82.6%) (Table 1), and most cases were diagnosed at an advanced stage, because 78% of women had FIGO stages III/IV. More than 95% of women had chemotherapy.

Women included in this study were divided according to histology of tumors. Continuous variables were compared by the Mann–Whitney test. Nominal and ordinal variables were compared by the chi-square test.

### 2.1. Differences in Fatty Acid Profile between Patients with EOC or Borderline Tumor

The fatty acid profiles of SSAT did not significantly differ between patients with EOC or borderline tumor (Table 2). Considering the limited number of participants, namely in borderline patients, the results do not show significant impact of aggressive tumors on adipose tissue composition, unlike the study by Yam and colleagues [25].

### 2.2. Correlation of PUFA Content among Different Abdominal Adipose Tissues

Among patients with EOC, 169 adipose tissue samples were harvested: 46 superficial subcutaneous adipose tissue (SSAT), 46 deep subcutaneous AT (DSAT), 38 visceral AT (VAT), and 39 omental AT (OAT).

We observed potential modifications of fatty acid profiles according to adipose tissue anatomic location (Table 3). We analyzed data for only women with adipose tissues retrieved from the four different anatomic locations, and fatty acid profiles were compared with SSAT as the reference. Whatever the adipose tissue, content of SFAs was ~28%, MUFAs about ~55%, and PUFAs, including n-6 and n-3 series, about 13%. The n-6 PUFAs were predominant as compared with PUFAn-3 because the ratio of PUFAn-6 to n-3 content was about 8.5.

Nevertheless, these different subclasses showed some modifications. To simplify, only modifications ≥10% of fatty acid rates are reported. As compared with SSAT, the DSAT, VAT, and OAT profiles showed an increase (+10 to +15%) in stearic acid (18:0). For SFAs, the major modifications were for long-chain SFAs (LC-SFAs), with increases of +55%, +66%, and +100% in DSAT, VAT, and OAT, respectively. In MUFAs, contents were little changed according to adipose locations: 10% increase in LC-MUFA content in DSAT. Conversely, DSAT, VAT, and OAT showed decreased long-chain PUFAn-6 (LC-PUFAn-6) content (−10 to −18%) as compared with SSAT. These decreases were linked to modifications in arachidonic acid (20:4n-6) content (−13% to −23%). Similarly, LC-PUFAn-3 content was decreased −17% in DSAT and −14% in OAT.

Taken together, these results showed a balance between LC-SFAs that increased in content and LC-PUFAs that decreased in content in deeper adipose tissues as compared with SSAT. Nevertheless, these changes especially occurred for LC fatty acids with low rates (<2%), which led to no significant difference in content of fatty acid subclasses (total SFAs, MUFAs, PUFAn-6, and PUFAn-6+n-3) between adipose tissues from different anatomic locations. However, total PUFAn-3 content showed a slight but significant decrease in DSAT and OAT as compared with SSAT.

We analyzed the correlation in fatty acid content between SSAT and other abdominal adipose tissues. Except for arachidonic acid (20:4n-6) in OAT, all fatty acids in SSAT showed good and significant correlation with fatty acids of DSAT, VAT, and OAT (Table 4). Nevertheless, all fatty acids combined, DSAT showed the best correlations with SSAT. The median correlation coefficient for DSAT was r^2^ = 0.87 and that of VAT and OAT was 0.79 and 0.70, respectively. The correlation for total PUFAn-6 and total PUFAn-6+n-3 content was r^2^ ≥ 0.90 between SSAT tissue and other abdominal adipose tissues.

### 2.3. Clinical Factors Associated with Recurrence-Free Survival with EOC

The median follow-up was 14.4 months (range 2.0–39.5) for the 46 patients with EOC, and 17 cases of OC recurrence were observed in this population. With the Kaplan–Meir method and log-rank test, several clinical parameters were associated with longer recurrence-free survival (RFS): overweight/obesity (*p* = 0.01), FIGO stage I-II (*p* = 0.06), absence of macroscopic tumor residue after surgery (*p =* 0.01), and absence of ascitic fluid at diagnosis (*p* = 0.009) (Figure 1).

### 2.4. High Level of PUFAs in Abdominal Tissues Associated with Longer RFS

The results of the association of fatty acid content in SSAT are provided in Figure 2 and in Appendix A for DSAT, VAT, and OAT, respectively. SFA and MUFA content were not significantly associated with RFS. Nevertheless, high PUFAn-6 content was associated, although sometimes not significantly, with longer RFS (SSAT *p* = 0.01 (Figure 2), DSAT *p* = 0.07 (Appendix A), VAT *p* = 0.01 (Appendix A), OAT *p* < 0.0004 (Appendix A)). High PUFAn-3 content was associated with longer RFS in subcutaneous adipose tissue (SSAT *p* = 0.01 (Figure 2) and DSAT *p* = 0.06 (Appendix A)) but not in VAT or OAT (EAT *p* = 0.53 (Appendix A), OAT *p* = 0.40 (Appendix A)). Finally, high content of total PUFA (n-6+n-3) was significantly associated with longer RFS whatever the considered abdominal adipose tissue (SSAT *p* < 0.0001 (Figure 2), DSAT *p* < 0.0001 (Appendix A), VAT *p* = 0.01 (Appendix A), OAT *p* = 0.0001 (Appendix A)). Median survival was 12 and 27 months for patients with low and high total PUFA content.

### 2.5. Polyunsaturated Fatty Acid Content Is an Independent Prognosis Factor for Recurrence-Free Survival

We present in Table 5 the results of the univariable analysis examining potential predictive factors for RFS in women with EOC. Age (*p* = 0.005); menopausal status (*p* = 0.04); advanced FIGO stage (*p* = 0.01); the performance of cytoreductive surgery; total long-chain polyunsaturated n-3 fatty acids level > median (*p* = 0.005); total long-chain polyunsaturated n-6 fatty acids level > median (*p* = 0.005); and total long-chain polyunsaturated (n-3+n-6) fatty acids level > median (*p* = 0.005) were eligible for multivariable analysis. For fatty acids, we chose to only consider total long-chain polyunsaturated (n-3+n-6) fatty acids level > median containing information on both subclasses.

The main finding of this analysis is that advanced FIGO stage, non-operated women, and total long-chain polyunsaturated fatty acids level were independent predictive factors of RFS in women with EOC

## 3. Discussion

Using paired adipose tissues from four locations in women with EOC, we found some subtle differences in fatty acid composition between subcutaneous and visceral fat tissues. Nevertheless, the SFA, MUFA, and PUFA subclasses remained equivalent in content, with strong correlations between PUFAn-6 and PUFAn-6+n-3 content in the four abdominal adipose tissues. Clinical factors associated with RFS were age, BMI, advanced FIGO stage, tumor residue after surgery, and total content of PUFA > median. Moreover, for the first time, our study showed an association between PUFA content and recurrence-free survival, and PUFA content of adipose tissue was an independent prognostic factor for EOC.

All adipose tissues of the body are exposed to the same lipid intake. Regarding fatty acid subclasses (SFAs, MUFAs, and PUFAs), our results showed similar fatty acid composition in the four analyzed adipose tissues. These results agree with two studies [30,31] comparing fatty acid content of subcutaneous and omental/visceral fat tissues. The strongest correlations among the four adipose tissues were for PUFAn-6 and total PUFA (n-6+n-3). These results agree with the literature: the highest correlations between dietary fatty acids and their content in adipose tissues were for PUFAn-6 or n-3 and to a lesser extent SFAs [26,32]. Nevertheless, we noticed subtle differences between tissues, especially an increase in LC-SFA content and a decrease in LC-PUFA content in deep versus superficial subcutaneous tissue. Quality and quantity of dietary fat may affect adipose tissue gene expression linked to fatty acid synthesis and desaturation, lipogenesis and lipolysis, and dietary PUFAn-3 content can reverse the SFA effect [33]. On correlation analysis, fatty acid content was correlated more in both subcutaneous fat tissues than both visceral fat tissues. Several studies highlighted differences between visceral and subcutaneous adipose tissues in terms of cellularity, metabolism, insulin sensitivity, adipokine secretion, or immunity infiltration [34,35,36]. The turnover of triglycerides is not strictly the same according to the physiological situation of the body (obesity or not) when visceral and subcutaneous tissues are considered [28]. These differences in fatty acid composition could highlight the adipose metabolism according to anatomic locations. With a median BMI of 26.6 kg/m^2^ in this study, some patients could present divergent fatty acid turnover between their subcutaneous and visceral fat tissues. In addition, we cannot exclude that the presence of cancer can activate a process of lipolysis, especially in the visceral adipose tissue, which could manage an activation of the renewal of fatty acids with a specific rhythm, independently of the obesity status. 

Factors associated with better prognosis in the literature are age, early stages, or the absence of tumor residue after surgery [11,37]. Overweight and obesity appear to be risk factors for developing OC [38], but high BMI seemed to be associated with longer RFS for EOC patients included in this study. The prognostic significance of adiposity is still debated in the literature. Some studies report obesity associated with poor relapse/survival [39], favorably influencing survival [40] or with no effect [13,41,42]. These contradictory results suggest some confounding factors. Overweight/obesity masking skeletal muscle loss (sarcopenic obesity) or undernutrition induced by the cancer could be possibilities [12,43,44].

Our study showed associations between low PUFAn-6 and total PUFA n-6+n-3 contents and RFS in OC patients. The first hypothesis to explain the low PUFAn-6+n-3 content in adipose tissue may be low PUFA intake and/or malnutrition. The best suppliers of PUFAs are plant and animal fat sources. Further analysis of patient diet using a food frequency questionnaire will be necessary to identify whether this deficiency originates from food intake. Nevertheless, this hypothesis is supported by the Playdon study [16]: the pre-diagnosis diet surveys revealed worse survival of OC patients with low PUFA/MUFA ratio intake. In addition, a severe malnutrition index is associated with poor RFS and overall survival [45,46], but a nutritional intervention can improve survival [46]. On the other hand, several studies [47], including the one we published on breast adipose tissue [48], show that n-6 and n-3 PUFA levels increase with age after menopause. These changes particularly concern long-chain PUFAs (>20 carbons). In this pilot study, four patients were not post-menopausal and could confound the interpretation of the results. Relapse-free survival analyses performed by excluding these four patients lead to identical conclusions with a significant decrease in RFS for patients with low PUFAn-6+3 levels. However, to overcome this factor of age, additional adipose fatty acid profiles will have to be acquired on non-menopausal patients.

The second hypothesis suggests the presence of the tumor, its metabolism, and its consequences on fluids and other tissues outsides the ovaries. Comparing OC patients to patients without cancer or with a benign tumor, Yam and colleagues reported a PUFAn-6 deficiency in adipose tissue of patients with OC and assumed PUFA consumption by the tumor [25]. Our results did not show different fatty acid profiles between EOC and borderline tumor patients. Limited number of borderline tumor patients may be a weakness of our study. Other studies also conclude that metabolism is turned to lipolysis in the context of ovarian cancer. Gercel-Taylor and colleagues showed that serum and ascites from ovarian cancer patients contain factors capable of inducing lipolysis of adipocytes [49]. Two studies associated low concentrations of LDLs and glycerophospholipids to unfavorable survival [23,24]. One may wonder why PUFAs n-6 and n-3 are particularly consumed. The entry and exit of fatty acids from adipose tissue have certain selectivity, depending on the length of the chains and the degree of unsaturation. MUFAs (namely 18:1n-9, oleic acid) and some PUFA n-6 are preferentially taken up by adipose tissue after a meal [50], but PUFA n-3 (C20:5n-3 and C18:3n-3) and PUFA n-6 (C20:4n-6 and C18:2n-6) are the first mobilized fatty acids during lipolysis [51]. A few studies have pointed to the possibility of specific elevated consumption of PUFAs by the tumor, which would balance an endogenous synthesis particularly oriented toward saturated and monounsaturated fatty acids. This observation was confirmed in colon cancer patients, who show reduced levels of circulating PUFAs [52] or in adipose tissues from patients with OC [25]. A recent study showed for the first time increased consumption of n-6 PUFAs and n-3 PUFAs in colonic cancer cells as compared with normal colonocytes [53]. This line of research deserves to be explored for OC.

The number of patients included in this pilot study may seem a limitation, and further studies are needed to determine whether PUFAs (n-6 and n-3) depletion originates food intake and/or in tumor metabolism. Nevertheless, the PUFAs (n-6+n-3) level is an independent prognostic marker in our study. These data may guide research avenues to design new therapeutic strategies for patients with ovarian cancer whose prognosis remains very poor.

## 4. Material and Methods

### 4.1. Study Population

The study was performed with approval of the Tours university teaching hospital review board. Written informed consent was obtained from all participants prior to the study. The patients involved in the present study are part of the METERMUS trial registered on clinicaltrials.gov (NCT03027479).

We included women with an invasive EOC or borderline ovarian tumor treated between March 2017 and January 2020 for whom adipose tissue samples were excised during surgery and stored in our university hospital cryobank. We excluded women with non-EOC, endometrial tumors, or tumors with a digestive tract origin.

A pathologist (F.A.) experienced in ovarian pathology performed histological analyses and reviewed the surgical excision specimens and biopsy slides and determined tumor phenotypes (borderline or: serous, endometrioid, clear-cell, mucinous, or other for OC). Combining physical examination, biopsy analysis, imaging, and intra-abdominal exploration during surgery or laparoscopy, cancer stage was determined with the FIGO system (International Federation of Gynecology and Obstetrics). The tumor size, location, and extension to nearby organs, spread to lymph nodes, or colonization of peritoneal cavity/metastases led to assigning an overall stage (stages I and II group localized cancers without lymph node invasion and stages III and IV cancers with nearby organ colonization, lymph node invasion, or metastases).

Using the computerized medical records in our institution, clinical characteristics obtained were age, menopausal status, and chronic pathologies (high blood pressure, diabetes, or hypercholesterolemia/dyslipidaemia). BMI measurement led to subdividing women into four groups according to the World Health Organization: underweight (BMI< 18.5 kg/m^2^), normal weight (BMI 18.5–24.9 kg/m^2^), overweight (BMI 25.0–29.9 kg/m^2^), and obese (BMI ≥ 30.0 kg/m^2^). During follow-up, the occurrence of relapse was recorded.

### 4.2. Sample Details

Adipose-tissue samples were obtained from women with EOC and borderline phenotype. If possible, samples were excised from four locations: superficial and deep subcutaneous (SSAT and DSAT), visceral (pericolic) (VAT), or omental (OAT). Superficial subcutaneous adipose tissue is between the skin and external oblique muscle and deep subcutaneous adipose tissue between transversal muscle fascia and parietal peritoneum. Epiploic appendages are fat-filled sacs along the surface of the colon. Omental samples were biopsied from omentum, the fatty apron that stretches over the intestines, liver, and stomach. Samples were stored in liquid nitrogen until analysis.

### 4.3. Fatty Acid Analysis

As described [54], total lipids were extracted from 50 mg frozen adipose tissue samples with the Folch method [55] with 8 mL of chloroform methanol (2:1) mixture. Thin layer chromatography (cat# 1057150001, Millipore, Guyancourt, France) was used to purify triglycerides, and transmethylation of fatty acids was performed using boron trifluoride (CAS #2802-68-8) in methanol solution (Sigma-Aldrich, Saint-Quentin Fallavier, France). Methylated fatty acids were analyzed by capillary gas chromatography (GC-2010 Plus chromatograph, Schimadzu, France) with a polar column (BPX70 column, 60 m x id 0.25 mm, cat#054623, SGE Analytical Science, Courtaboeuf, France), and a standard mixture permitted identification (Supelco 37 Component FAME Mix, cat#CRM47885, Merck, Molsheim, France). Fatty acid levels are expressed as percentage of the total integrated area. Chromatography analysis was supervised and validated by an experienced analytical biochemist (C.G.) with blinding to clinical data. In this study, fatty acids of 20 to 24 carbons were considered long-chain fatty acids.

### 4.4. Statistical Analysis

Statistical analyses were performed with R 2.13.1 (http://www.cran.r-project.org/ (accessed on 2 November 2022)) and GraphPad Prism 6.01 (GraphPad Software, Inc, San Diego, CA, USA). Demographic and baseline clinical characteristics were summarized with medians and interquartile range, IQR) for continuous variables and number (percentages) for ordinal and nominal variables. The chi-square test was used to compare ordinal and nominal data, Mann–Whitney test for continuous data. To compare adipose tissue composition from 4 abdominal locations, nonparametric Friedman’s ANOVA test for repeated measures, with Dunn’s post test was used and Spearman test for correlations. Survival analysis involved the Kaplan–Meier method with the log-rank test. For these analyses, the participants were separated into two groups according to fatty acid content (< or ≥ median); *p* < 0.05 was considered statistically significant.

To identify RFS predictive factors, patients’ data were tested individually for association with RFS using log-rank tests. Factors with a *p*-value <0.1 were then included in a Cox proportional hazards model. Multivariate analyses were performed by using the Cox proportional hazards model. Hazard ratios were calculated from the model coefficients; *p* < 0.05 was considered statistically significant.

## Figures and Tables

**Figure 1 ijms-24-01768-f001:**
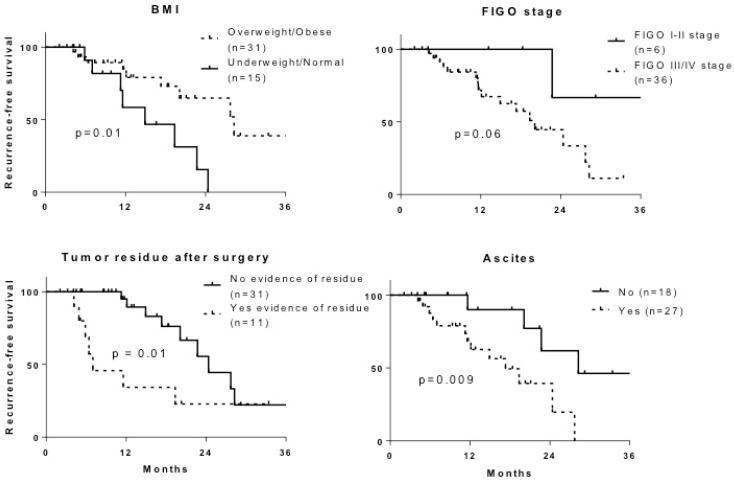
Recurrence-free-survival (RFS) of patients with epithelial ovarian cancer (EOC) by age, body mass index, or tumor extension. Analysis of recurrence-free survival with the Kaplan–Meier method and log-rank test.

**Figure 2 ijms-24-01768-f002:**
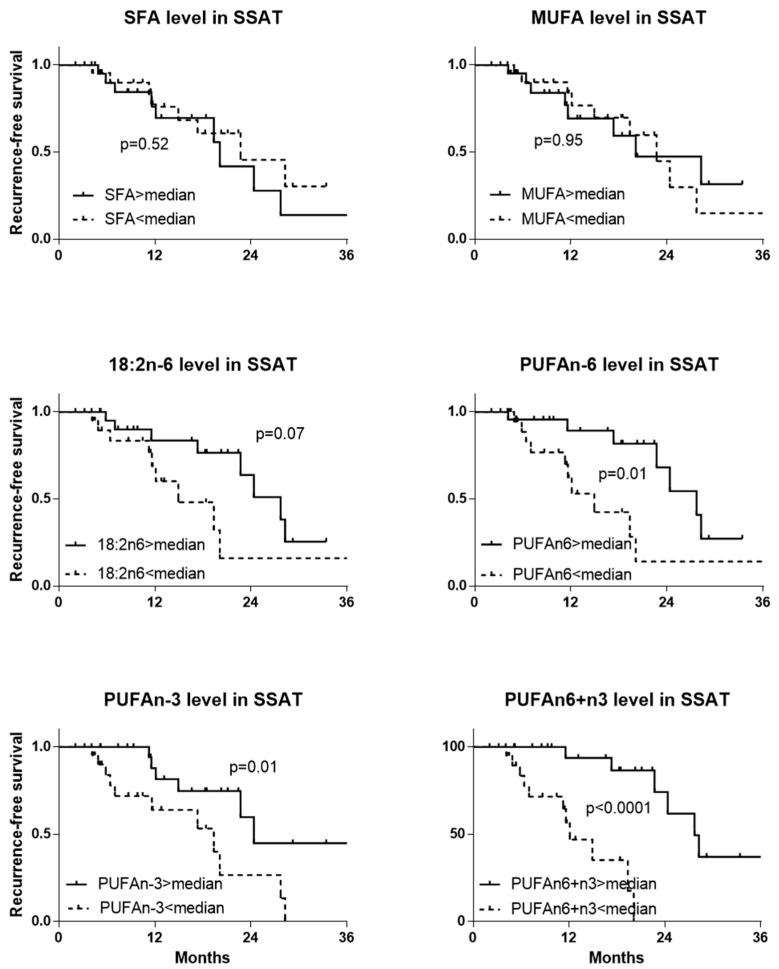
Recurrence-free survival in patients with EOC by fatty acid content in superficial subcutaneous adipose tissue. Analysis of recurrence-free survival with the Kaplan–Meier method and log-rank test. Patients were divided into two groups by fatty acid content (<median, *n* = 23 or >median, *n* = 23) (*n* = 46). SFAs: saturated fatty acids, MUFAs: monounsaturated fatty acids, PUFAs: polyunsaturated fatty acids, SSAT: superficial subcutaneous adipose tissue, 18:2n-6: linoleic acid.

**Table 1 ijms-24-01768-t001:** Demographic and histological characteristics of women included in the study.

	Ovarian Cancer *n* = 46		Borderline Tumors *n* = 6		*p*
	Median or *n* (%)	[IQR]	Median or *n* (%)	[IQR]	
Age (years)	69.0	[61.0–74.3]	64.0	[50.3–71.3]	0.34
Weight (kg)	65.5	[59.0–73.5]	65.0	[59.0–93.5]	0.76
BMI (kg/m^2^)	26.6	[23.3–29.8]	25.8	[22.3–38.1]	0.81
- Underweight	2 (4.3%)		0 (0%)		0.60
- Normal weight	13 (28.7%)		3 (50.0%)		
- Overweight	18 (39.1%)		1 (16.7%)		
- Obese	13 (28.3%)		2 (33.3%)		
No. of children	2	[2–3]	2	[1.5–3.5]	0.92
Menopause	42 (91.3%)		5 (83.3%)		0.47
Hormonal treatment	11 (23.9%)		0 (0%)		0.32
HBP	16 (34.8%)		3 (50.0%)		0.65
Diabetes	4 (8.7%)		2 (33.3%)		0.13
Dyslipidaemia	9 (19.6%)		3 (50.0%)		0.12
Histology					
- Serous carcinoma	38 (82.6%)				
- Endometroid	2 (4.3%)				
- Clear cell	1 (2.2%)				
- Mucinous	1 (2.2%)				
- Mixed	3 (6.5%)				
- Other	1 (2.2%)				
FIGO stage					
- I	5 (10.9%)				
- II	1 (2.2%)				
- III	26 (56.5%)				
- IV	10 (21.7%)				
- Unknown	4 (8.7%)				
Chemotherapy	44 (95.7%)				

IQR: interquartile range, BMI: body mass index, HBP: high blood pressure. BMI: underweight (BMI< 18.5 kg/m^2^), normal weight (BMI 18.5–24.9 kg/m^2^), overweight (BMI 25.0–29.9 kg/m^2^), and obese (BMI ≥ 30.0 kg/m^2^). FIGO: International Federation of Gynecology and Obstetrics.

**Table 2 ijms-24-01768-t002:** Fatty acid composition of superficial subcutaneous adipose tissue of patients.

	Ovarian Cancer *n* = 46		Borderline Tumor *n* = 6		*p*
Fatty Acids ^a^	Median	IQR	Median	IQR	
SFAs					
14:0	2.97	2.57–3.23	3.03	2.56–3.19	0.94
16:0	20.62	19.56–21.91	19.9	18.17–21.84	0.37
18:0	3.57	3.06–4.90	4.32	2.21–5.26	0.83
LC Sat	0.17	0.14–0.28	0.17	0.13–0.38	0.96
Total SFAs	28.32	26.79–29.61	28.04	23.19–32.14	0.79
MUFAs					
14:1	0.36	0.27–0.46	0.30	0.29–0.43	0.75
16:1	4.59	3.56–5.51	4.12	3.44–5.04	0.64
18:1n-9c	46.67	45.37–48.94	46.98	44.14–49.21	0.79
18:1n-7c	2.02	1.82–2.23	2.14	1.89–2.36	0.40
LC mono	0.86	0.72–1.00	0.83	0.78–1.09	0.70
Total MUFAs	55.33	52.85–57.12	53.94	51.33–58.28	0.64
PUFAs					
18:2n-6c	9.73	8.90–11.62	10.76	9.70–12.60	0.27
18:3n-6	0.06	0.04–0.07	0.05	0.05–0.08	0.99
20:4n-6	0.42	0.32–0.57	0.44	0.21–0.55	0.58
LC n-6	1.30	0.93–1.55	1.04	0.80–1.57	0.47
Total PUFAn-6	10.97	10.08–12.98	11.74	10.61–14.53	0.37
18:3n-3	0.61	0.50–0.77	0.73	0.58–0.97	0.21
20:5n-3	0.08	0.06–0.10	0.06	0.04–0.13	0.53
22:5n-3	0.35	0.27–0.47	0.32	0.22–0.42	0.38
22:6n-3	0.24	0.17–0.33	0.20	0.14–0.32	0.44
LC n-3	0.73	0.60–0.92	0.66	0.51–0.81	0.46
Total PUFAn-3	1.41	1.10–1.61	1.45	1.16–1.88	0.62
Ratio PUFAn-6/n-3	8.27	7.25–9.86	9.15	5.99–12.03	0.79
PUFA n-6 + n-3	12.56	11.27–14.47	13.08	12.15–15.86	0.25

^a^ Fatty acids are expressed as % of total integrated area. IQR: interquartile range; SFAs: saturated fatty acids, MUFAs: monounsaturated fatty acids, PUFAs: polyunsaturated fatty acids, LC: long-chain fatty acids (20 to 24 carbons). Comparison is by the Mann–Whitney test.

**Table 3 ijms-24-01768-t003:** Comparison of fatty acid profiles between the four abdominal adipose tissue locations in 34 EOC patients (with two subcutaneous and two visceral samples).

	SSAT		DSAT		VAT		OAT	
Fatty Acids ^a^	Median	IQR	Median	IQR	Median	IQR	Median	IQR
14:0	2.97	2.51–3.19	3.11	2.59–3.32 **	3.06	2.68–3.41 *	3.14	2.68–3.50 ****
16:0	20.45	19.15–21.95	20.37	19.10–21.77	20.64	18.66–21.58	20.63	18.79–21.39
18:0	3.57	3.06–4.70	**↗** 4.00	3.57–4.70 *	**↗** 4.10	3.48–4.90 ***	**↗** 3.94	3.50–5.16 ****
LC-SFAs	0.18	0.14–0.28	**↗** 0.28	0.21–0.36 **	**↗** 0.30	0.23–0.37 ****	**↗** 0.36	0.25–0.43 ****
Total SFAs	28.03	26.38–29.47	28.84	25.91–30.11	28.91	26.54–30.25	28.86	27.17–30.07
14:1	0.35	0.26–0.46	**↘** 0.30	0.23–0.41 ***	0.34	0.24–0.45	0.39	0.26–0.46
16:1	4.61	3.56–5.53	4.57	3.35–5.76	4.29	3.47–5.57	4.70	3.97–5.74
18:1n-9c	46.71	45.72–48.78	46.62	45.43–48.65	46.46	45.30–48.19	46.33	45.48–47.80
18:1n-7c	2.08	1.87–2.30	2.07	1.91–2.35	2.02	1.81–2.13 *	1.93	1.77–2.17 **
LC-MUFAs	0.87	0.75–1.07	**↗** 0.96	0.84–1.11 **	0.86	0.76–1.10	0.95	0.88–1.06 ***
Total MUFAs	55.36	52.94–57.12	55.20	52.70–57.46	55.45	52.67–56.64	55.49	53.05–56.46
18:2n-6c	9.74	8.90–11.90	9.79	8.73–12.00	10.06	8.77–12.14	10.02	8.73–11.81
18:3n-6	0.05	0.04–0.07	0.05	0.04–0.07	0.05	0.04–0.07	0.06	0.05–0.08
20:4n-6	0.43	0.33–0.57	**↘** 0.37	0.26–0.44 **	**↘** 0.33	0.28–0.41 ***	**↘** 0.36	0.27–0.50 ***
LC-PUFAn-6	1.32	1.01–1.55	**↘** 1.18	0.90–1.43	**↘** 1.12	0.93–1.32 ***	**↘** 1.08	0.80–1.28 ***
Total PUFAn-6	11.39	10.08–13.00	11.26	9.77–13.14	11.14	9.90–13.05	11.27	9.94–12.67 **
18:3n-3	0.64	0.52–0.79	0.59	0.46–0.72 ***	0.58	0.47–0.78	0.60	0.50–0.76
20:5n-3	0.07	0.06–0.09	**↘** 0.06	0.05–0.07 ***	**↘** 0.06	0.05–0.08 *	**↘** 0.06	0.05–0.09
22:5n-3	0.35	0.27–0.47	**↘** 0.30	0.24–0.41 **	0.35	0.27–0.43	**↘** 0.29	0.23–0.40 ****
22:6n-3	0.23	0.17–0.31	**↘** 0.18	0.13–0.27 ***	0.25	0.20–0.30	0.22	0.15–0.33
LC-PUFAn-3	0.71	0.61–0.89	**↘** 0.59	0.49–0.87 ***	0.72	0.58–0.85	**↘** 0.61	0.49–0.81 ***
Total PUFAn-3	1.41	1.11–1.55	↘ 1.23	1.02–1.39 ****	1.31	1.10–1.55	1.29	1.09–1.44 ***
PUFA n-6+n-3	12.90	11.24–14.57	12.51	10.77–14.31 **	12.64	11.0–14.92 *	12.62	11.07–13.92 ****
Ratio n-6/n-3	8.22	5.30–15.40	**↗** 9.31	5.68–17.53 ****	8.79	4.83–15.08	8.90	4.43–17.65 ***

^a^ Fatty acids were expressed as % of integrated area, IQR: interquartile range, SFAs: saturated fatty acids, MUFAs: monounsaturated fatty acids, PUFAs: polyunsaturated fatty acids, LC: long-chain fatty acids (from 20 to 24 carbons). SSAT: superficial subcutaneous adipose tissue, DSAT: deep subcutaneous adipose tissue, VAT: visceral adipose tissue, OAT: omental adipose tissue. Only women with adipose tissues from the four different anatomic locations were included in this table (34 with OC). Difference between compositions of the 4 adipose tissues were analyzed using the nonparametric Friedman’s ANOVA test for repeated measures, and DSAT, VAT, and OAT were compared to SSAT with Dunn’s post test. * *p* ≤ 0.05, ** *p* ≤ 0.01, *** *p* ≤ 0.001, **** *p* ≤ 0.0001. Arrows (↗ or ↘) were added to highlight the fatty acid variations ≥10% (increase or decrease) when significant differences were observed.

**Table 4 ijms-24-01768-t004:** Spearman correlation between fatty acid content in superficial subcutaneous adipose tissue versus other abdominal adipose tissues in 34 EOC patients (with two subcutaneous and two visceral samples).

	SSAT versus					
	DSAT		VAT		OAT	
Fatty Acids	r^2^	*p*	r^2^	*p*	r^2^	*p*
14:0	**0.93**	****	**0.86**	****	**0.84**	****
16:0	**0.87**	****	0.79	****	0.60	***
18:0	**0.84**	****	**0.80**	****	0.73	****
LC Sat	0.74	****	0.67	****	0.54	***
Total SFAs	0.78	****	0.68	****	0.50	**
14:1	**0.90**	****	**0.89**	****	0.77	****
16:1	**0.82**	****	0.75	****	0.72	****
18:1n-9c	**0.90**	****	**0.88**	****	0.78	****
18:1n-7c	0.79	****	0.63	****	0.54	***
LC mono	**0.81**	****	0.76	****	0.57	***
Total MUFAs	**0.85**	****	0.78	****	0.64	****
18:2n-6c	**0.98**	****	**0.98**	****	**0.92**	****
18:3n-6	**0.82**	****	0.76	****	0.69	****
20:4n-6	**0.88**	****	0.52	**	0.42	*
LC n-6	**0.87**	****	0.73	****	0.60	***
Total PUFAn-6	**0.96**	****	**0.96**	****	**0.92**	****
18:3n-3	**0.91**	****	**0.91**	****	**0.87**	****
20:5n-3	**0.82**	****	0.77	****	0.56	***
22:5n-3	**0.90**	****	0.77	****	0.72	****
22:6n-3	**0.91**	****	**0.82**	****	**0.80**	****
LC n-3	**0.92**	****	0.74	****	0.76	****
Total PUFAn-3	**0.91**	****	**0.89**	****	**0.82**	****
PUFA n-6+n-3	**0.96**	****	**0.96**	****	**0.90**	****

Only women with adipose tissues from the four different anatomic locations were included in this table (34 with OC). Content of each fatty acid in SSAT was compared to content of the same fatty acid in DSAT, VAT, and OAT by the Spearman correlation test. * *p* ≤ 0.05, ** *p* ≤ 0.01, *** *p* ≤ 0.001, **** *p* ≤ 0.0001. To highlight the best correlation coefficients, they are written in bold (r^2^ ≥ 0.80) and red (r^2^ ≥ 0.90). SFAs: saturated fatty acids, MUFAs: monounsaturated fatty acids, PUFAs: polyunsaturated fatty acids, LC: long-chain fatty acids (20 to 24 carbons). SSAT: superficial subcutaneous adipose tissue, DSAT: deep subcutaneous adipose tissue, VAT: visceral adipose tissue, OAT: omental adipose tissue.

**Table 5 ijms-24-01768-t005:** Predictive factors for recurrence free survival in women with EOC.

	Univariate Analysis		Multivariate Analysis	
	HR [IC 95%]	*p*-Value	HR [IC 95%]	*p*-Value
Age	0.93 [0.90–0.97]	**0.005**	-	
Menopause	0.27 [0.07–0.98]	**0.04**	0.12 [0.003–4.28]	0.24
BMI	0.92 [0.82–1.04]	0.19	-	
FIGO Stage				
Early stages (I–II)	Reference			
Advanced stages (III–VI)	7.1 [1.52–33.3]	**0.01**	2.35 [0.92–5.98]	0.05
Residue after surgery			-	
No residue	Reference			
Microscopic residue	0.83 [0.09–7.3]	0.86	6.5 [0.07–6.32]	0.71
No surgery	4.04 [0.77–21.1]	**0.09**	17.9 [1.5–240]	0.03
Serum albumin	1.04 [0.91–1.19]	0.57		
Myristic acid >med	2.07 [0.64–6.62]	0.22	-	
Palmitic acid > med	1.96 [0.66–5.80]	0.23	-	
Stearic acid > med	0.97 [0.35–2.68]	0.95		
Total SFA > med	2.56 [0.84–7.79]	0.10		
Myristoleic acid > med	3.61 [1.09–11.9]	0.04		
Palmitoleic acid > med	1.17 [0.41–3.34]	0.76		
Oleic acid (OA) > med	0.81 [0.29–2.25]	0.69		
Vaccenic acid > med	2.87 [0.86–9.49]	0.08		
Total MUFA > med	0.82 [0.29–2.34]	0.71		
Linoleic acid (LA) >med	0.28 [0.08–0.97]	0.04		
Gamma Linolenic acid > med	0.97 [0.35–2.68]	0.95		
Arachidonic acid > med	0.62 [0.21–1.83]	0.39		
Total LC n-6 > med	0.59 [0.20–1.73]	0.33		
Total PUFAn-6 > med	0.11 [0.03–0.52]	**0.005**		
Alpha Linolenic acid > med	0.58 [0.21–1.65]	0.31		
Eicosapentaenoic acid > med	0.44 [0.16–1.25]	0.12		
Docosapentaenoic acid > med	0.53 [0.19–1.51]	0.24		
Docosahexaenoic acid > med	0.60 [0.21–1.66]	0.32		
Total LC n-3 > med	0.91 [0.33–2.55]	0.86		
Total PUFAn-3 > med	0.31 [0.11–0.93]	**0.03**		
Total PUFA > med	0.11 [0.02–0.52]	**0.005**	0.01 [0.0001–0.76]	**0.01**

Med: median, BMI: body mass index. FIGO: International Federation of Gynecology and Obstetrics. Factors with *p*-value in bold were evaluated in the multivariate analysis.

## Data Availability

Not applicable.

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
