# Peer review of "Total Polyunsaturated Fatty Acid Level in Abdominal Adipose Tissue as an Independent Predictor of Recurrence-Free Survival in Women with Ovarian Cancer"

_ijms, 2023, doi:10.3390/ijms24021768_

Round 1

Reviewer 1 Report

The topic of the manuscript written by Salaun and colleagues is interesting since the high incidence and high risk of recurrence in woman with ovarian cancer. I would like to congratulate with the authors for the job done. However, authors should present better the results and clarify some points. Below are a few items for consideration.

Abstract:

·        Authors should add “~” before the numbers indicating the percentages of the different types of fatty acids.

·        “For EOC patients, median follow-up was 15 months” and “EOC patients were separated into two groups by median fatty acid content” should be moved to the methods section of the abstract. In addition, authors should explain better what type of analysis they did with those two groups.

·        “FIGO stage, tumor residue after surgery and body mass index, were clinical predictors of RFS” should be moved to the background section or removed.

Line 155: range should be changed in IQR.

Lines 155-157 (and throughout the manuscript): categorical variables should be changed in ordinal and nominal variables.

In the 2.4 section authors should describe what kind of test they used to identify the predictive factors (table 5).

Table 1: range should be changed in IQR.

Linea 180-181: In my opinion, considering the limited numbers of participants enrolled, authors can’t state that aggressive tumors do not regulate fatty acid content in adipose tissue. This sentence need reformulation.

Table 2: I think that authors shown median and not mean values, so they should change mean in median.

Lines 194-195: Authors should add “~” before the numbers indicating the percentages of the different types of fatty acids

Line 206: Friedman test not compare in statistical analysis section

Line 226: authors should indicate what EAT means.

Section 3.5: authors should describe in the text the values of OR reported in the table. Under the table authors should specify when a p value is considered statistically significant and explain why some p-value are in bold. In addition, why authors didn’t add in the analysis the other comorbidities shown in table 1 (es. Diabetes, HBP ecc.)

Figure 2 and supplementary figures: authors should indicate in the figures the numbers of participants of each group.

Line 355: authors should specify that they are talking about PUFAs levels in adipose tissue

Author Response

Abstract:

  • Authors should add “~” before the numbers indicating the percentages of the different types of fatty acids.

This has been done

  • “For EOC patients, median follow-up was 15 months” and “EOC patients were separated into two groups by median fatty acid content” should be moved to the methods section of the abstract. In addition, authors should explain better what type of analysis they did with those two groups.

This have been done

“Fatty acids composition were compared between the 4 locations usingnon-parametric Friedman's ANOVA test for repeated measures, with Dunn’s post test was used on repeated measures (paired Friedman test with Dunn’s post test). Median follow-up of EOC patients was 15 monthsand patients RFS was analyzed using Kaplan-Meier survival curves and Log Rank test by separating patients into 2 groups according to median fatty acids levels.”

  • “FIGO stage, tumor residue after surgery and body mass index, were clinical predictors of RFS” should be moved to the background section or removed.

It has been moved to the background section

Line 155: range should be changed in IQR.

Range has been changed in IQR (line 155) and numbers in Table 1, Table 2, and Table 3.

Lines 155-157 (and throughout the manuscript): categorical variables should be changed in ordinal and nominal variables.

This has been done

In the 2.4 section authors should describe what kind of test they used to identify the predictive factors (table 5).

As requested by the reviewer we added:

To identify RFS predictive factors, patient’s data were tested individually for association with RFS using log-rank tests. Factors with a p value <0.1 were then included in a Cox proportional hazards model. Multivariate analyses were performed by using the Cox proportional hazards model. Hazard ratios were calculated from the model coefficients.

Table 1: range should be changed in IQR.

This has been done

Linea 180-181: In my opinion, considering the limited numbers of participants enrolled, authors can’t state that aggressive tumors do not regulate fatty acid content in adipose tissue. This sentence need reformulation.

As requested by the reviewer, we made the following change:

Considering the limited number of participants, namely in borderline patients, the results do not show an impact of aggressive tumors on adipose tissue composition, unlike the study by Yam 1997.

Table 2: I think that authors shown median and not mean values, so they should change mean in median.

The reviewer is right and we apologize for this unfortunate error.

Lines 194-195: Authors should add “~” before the numbers indicating the percentages of the different types of fatty acids

This has been done

Line 206: Friedman test not compare in statistical analysis section

The reviewer is right and we changed the sentence by :

“Difference between compositions of the 4 adipose tissues were analyzed using a non-parametric Friedman's ANOVA test for repeated measures, and DSAT, VAT and OAT were compared to SSAT with Dunn’s post test”.

Line 226: authors should indicate what EAT means.

The reviewer is correct, we apologize for this error, EAT (epiploic) was an earlier abbreviation that we replaced with VAT (visceral) in the final manuscript. The abbreviation EAT was deleted on line 226, and it was replaced by VAT in Suppl Fig S1B.

Section 3.5: authors should describe in the text the values of OR reported in the table. Under the table authors should specify when a p value is considered statistically significant and explain why some p-value are in bold. In addition, why authors didn’t add in the analysis the other comorbidities shown in table 1 (es. Diabetes, HBP ecc.)

We totally agree the comment of the reviewer and we added explanations in the « material &methods section » and in the table footnotes.

no predictive model including all these factors is available to help physicians faced to the patients with OC. Multivariate analysis tend to outperform both expert clinicans and predictive instruments based on risk grouping. OC prognosis is essentially based on the occurrence of relapses. However, international guidelines remain somewhat blurred concerning those women with increased risk of recurrence (as factors are considered separately) and there is lack of information on individualized factors for these patients. The current model is the first personalised tool based on predictive model which deal with the question of the association of fatty acid composition and RFS in OC patients

Other factors as (es. Diabetes, HBP ecc.) are already known as non associated with OC prognosis

Figure 2 and supplementary figures: authors should indicate in the figures the numbers of participants of each group.

This has been done

Line 355: authors should specify that they are talking about PUFAs levels in adipose tissue

We have added PUFAs (n-6+n-3)

We have directly addressed the reviewer’s concerns. These suggestions were extremely helpful and have substantially improved the overall clarity of the manuscript.

All the requested additions/changes are highlighted in yellow in the manuscript.

Reviewer 2 Report

This is an interesting work but major information is missing to conclude.

-The introduction is rich in information but it is far from your real subject, it is necessary to value the fatty acid part and its metabolism in adipose tissue in general and especially in relation to ovarian cancer.

- You based your work on non-significant results reported by the pleydon study.

« In this last Playdon study (19), nutrient analysis of lipid quality details found high content of saturated fatty acids (SFAs) associated but not significantly with increased mortality (p=0.07) and high polyunsaturated/monounsaturated fatty acid (PUFA/MUFA) ratio with increased survival (p=0.06) »

- the number of control cases (borderline tumor patients) is too low to have a significant comparison and a net effect.

- How we can speak in the discussion part about obesity and poor nutrition! Normally with poor nutrition, we will not have obesity!

Some studies report obesity associated with poor relapse/survival (41), favourably influencing survival (42) or with no effect (16,43,44). These contradictory results suggest some confounding factors. Overweight/obesity masking skeletal muscle loss or poor nutrition could be possibilities (15,45)”.

- How can we control the consumption of the PUFAn-6 and PUFAn-3 of our cohort? 

- In the discussion, you need to explain and talk more about the regulation of intake fatty acid metabolism and the composition of adipose tissue especially when we change the types of these fatty acids.

- In your discussion, you said that 

Triglyceride age (about 2 years) was relatively stable in visceral tissue whatever the BMI (only increased with morbid obesity) but was increased in subcutaneous fat tissue by 0.6 years when patients were overweight or obese” 

But in the “Clinical factors associated with recurrence-free survival with EOC 

The median follow-up was 14.4 months (range 2.0-39.5) for the 46 patients with EOC, and 17 cases of OC recurrence were observed in this population.”

I think you have to discard the cases that are not included in the age range of triglyceride stability.

- We know that adipose tissue composition is influenced by menopausal status. So how you discarded this parameter? 

The idea of the study is very good but you need to broaden your discussion approach further and increasingly focus on the relationship between the fatty acid composition of adipose tissue and cancer.

All these remarks must be taken in consideration. Answers to these criticisms will reinforce the quality of the manuscript and will permit to conclude with more accuracy.

Author Response

Comments and Suggestions for Authors

This is an interesting work but major information is missing to conclude.

-The introduction is rich in information but it is far from your real subject, it is necessary to value the fatty acid part and its metabolism in adipose tissue in general and especially in relation to ovarian cancer.

Reviewer is right. We have shortened the section on ovarian cancer in general and we have expanded the section on adipose tissue and ovarian cancer.

“Several recent studies indicate an alteration of lipid metabolism in ovarian cancer cells, with an increase of endogenous lipid synthesis and fatty acid absorption by tumor cells to meet their high energy demand (22,23). The relationship between ovarian cancer cells and visceral adipose tissue is a typical feature of ovarian cancer, which induces metabolic adaptation of the tumor to a lipid-rich environment and promotes ascites for-mation and metastasis. Reducing fatty acid import by neutralizing CD36 or FABP4 transporters in tumor cells controls tumor progression, metastasis and increases sensitiv-ity to chemotherapy (Mukherjee 2018, Gharpure 2018). Future therapeutic strategies may target lipid metabolism Chen 2019, Lui 2017 (24–26) but exploration of fatty acid metabo-lism remains uncommon in ovarian cancer patients. A reduction of plasma LDL and polyunsaturated glycerophospholipids in ovarian cancer patients with shorter survival (Bahmayr 2017, Zhu 2018). We noticed only one article which reported consequences ovarian cancer on adipose tissue composition with a mobilization of linoleic acid from subcutaneous and omental adipose tissues (Yam 1997). The fatty acid composition of ad-ipose-tissue fatty acids is considered to reflect dietary habits. A number of studies pro-posed adipose composition as a relevant biomarker to assess past lipid intake quality. In-deed, it reflects lipid intake during several months because of the slow turnover of fatty acids which half-life have been estimated between 1 and 2 years in healthy people with stable weight (Strawford 2004, Spalding 2017). However, this turnover is dynamic in order to adapt to pathophysiological changes in the body, starting with weight loss or gain. Thus, the greater the lipolysis, the faster the turnover, and vice versa. (Spalding 2017, Arner 2011). We hypothesized that fatty acid composition of adipose tissue may be impacted by the strong lipid uptake and fatty acid oxidation of ovarian cancer cells et adipose fatty ac-id profiles may be associated with outcomes in women with EOC.”

- You based your work on non-significant results reported by the pleydon study.

« In this last Playdon study (19), nutrient analysis of lipid quality details found high content of saturated fatty acids (SFAs) associated but not significantly with increased mortality (p=0.07) and high polyunsaturated/monounsaturated fatty acid (PUFA/MUFA) ratio with increased survival (p=0.06) »

This reference has been removed during the restructuring of the introduction

- the number of control cases (borderline tumor patients) is too low to have a significant comparison and a net effect.

The reviewer is right and we propose to conclude directly in the results section with the following sentence:

“Considering the limited number of participants, namely in borderline patients, the results do not show significant impact of aggressive tumors on adipose tissue composition, unlike the study by Yam 1997.”

- How we can speak in the discussion part about obesity and poor nutrition! Normally with poor nutrition, we will not have obesity!

“Some studies report obesity associated with poor relapse/survival (41), favourably influencing survival (42) or with no effect (16,43,44). These contradictory results suggest some confounding factors. Overweight/obesity masking skeletal muscle loss or poor nutrition could be possibilities (15,45)”.

We apologize for the poor choice of word. From a clinical point of view, the nutritional management of overweight or obese cancer patients is complex. Indeed, these patients may have a "hidden" undernutrition induced by the cancer that cannot be detected by the current definition criteria.The use of simple anthropometric measures may not provide information on body composition alterations, especially on the reduction of muscle mass, which may occur regardless of weight loss or BMI in cancer patients. Low muscle mass evaluation in overweight or obese cancer patient is still a challenging task.

We propose to introduce the expression “sarcopenic obesity” in the sentence, replace “poor nutrition” with “undernutrition induced by cancer”, and add the reference from Pardo et al 2016 -Sarcopenia and cachexia in the era of obesity: clinical and nutritional impact.

“These contradictory results suggest some confounding factors. Overweight/obesity masking skeletal muscle loss (sarcopenic obesity) or undernutrition induced by the cancer could be possibilities (15,45)+ Prado 2016”.

- How can we control the consumption of the PUFAn-6 and PUFAn-3 of our cohort? 

These data are the result of an ancillary study and the dietary questionnaire was not originally intended. If we assume that ovarian cancer development impacts whole-body lipid metabolism with the induction of lipolysis, adipose tissue can no longer be considered a biomarker of diet but a reflection of metabolic and nutritional status, and future studies should include a dietary questionnaire to estimate dietary PUFA intake.

In the discussion section, we propose to add the sentence below:

“Further analysis of patient diet using food frequency questionnaire will be necessary to identify whether this deficiency originates from food intake.”

- In the discussion, you need to explain and talk more about the regulation of intake fatty acid metabolism and the composition of adipose tissue especially when we change the types of these fatty acids.

The reviewer is right and we propose to add the paragraph below:

“One may wonder why PUFAs n-6 and n-3 are particularly consumed. The entry and exit of fatty acids from adipose tissue have a certain selectivity, depending on the length of the chains and the degree of unsaturation. MUFAs (namely 18:1n-9, oleic acid) and some PUFA n-6 are preferentially taken up by adipose tissue after a meal (Summers 2000), but PUFA n-3 (C20:5n-3 and C18:3n-3) and PUFA n-6 (C20:4n-6 and C18:2n-6) are the first mobilized fatty acids during lipolysis (Raclot 1997).In addition, a few studies have pointed to the possibility of specific elevated consumption of PUFAs by the tumor,which would balance an endogenous synthesis particularly oriented towards saturated and monounsaturated fatty acids.”

- In your discussion, you said that 

“Triglyceride age (about 2 years) was relatively stable in visceral tissue whatever the BMI (only increased with morbid obesity) but was increased in subcutaneous fat tissue by 0.6 years when patients were overweight or obese” 

But in the “Clinical factors associated with recurrence-free survival with EOC The median follow-up was 14.4 months (range 2.0-39.5) for the 46 patients with EOC, and 17 cases of OC recurrence were observed in this population.”

I think you have to discard the cases that are not included in the age range of triglyceride stability.

This is a good point. The first sentence was to explain and illustrate the differences in composition and metabolism between superficial vs visceral adipose tissues. However, these precise turnover dating are only valid for healthy individuals with stable weight whereas our patients have cancer with potentially significant lipid mobilization. For these reasons, we believe that it is preferable to remove these precise dating and focus on dynamic fatty acid turnover adapted to physiological conditions. As you suggest in the previous comment, we included in the introductory and discussion sections the motion of dynamic turnover of fatty acids adapting to pathophysiological conditions (weight gain or loss). Thus, the greater the lipolysis, the faster the turnover, and vice versa. (Spalding 2017, Arner 2011).The half-life of fatty acids estimated between 1 and 2 years, according to the studies, may be revised in ovarian cancer context.

For these reasons, the discussion paragraph has been completely restructured and we propose:

“All adipose tissues of the body are exposed to the same lipid intake. Regarding fatty acid subclasses (SFAs, MUFAs and PUFAs), our results showed similar fatty acid composition in the four analyzed adipose tissues. These results agree with two studies (32,33) comparing fatty acid content of subcutaneous and omental/visceral fat tissues. The strongest correlations among the four adipose tissues were for PUFAn-6 and total PUFA (n-6+n-3). These results agree with the literature: the highest correlations between dietary fatty acids and their content in adipose tissues were for PUFAn-6/n-3 and to a lesser extent SFAs (36,37). Nevertheless, we noticed subtle differences between tissues, especially an increase in LC-SFA content and a decrease in LC-PUFA content in deep versus superficial subcutaneous tissue. Quality and quantity of dietary fat may affect adipose tissue gene expression linked to fatty acid synthesis and desaturation, lipogenesis and lipolysis, and dietary PUFAn-3 content can reverse the SFA effect (34). On correlation analysis, fatty acid content was correlated more in both subcutaneous fat tissues than both visceral fat tissues. Several studies highlighted differences between visceral and subcutaneous adipose tissues in terms of cellularity, metabolism, insulin sensitivity, adipokine secretion or immunity infiltration (29–31). The turnover of triglycerides is not strictly the same according to the physiological situation of the body (obesity or not) when visceral and subcutaneous tissues are considered (35). These differences in fatty acid composition could highlight the adipose metabolism according to anatomic locations. With a median BMI of 26.6 kg/m² in this study, some patients could present divergent fatty acid turnover between their subcutaneous and visceral fat tissues. In addition, we cannot exclude that the presence of cancer can activate a process of lipolysis, especially in the visceral adipose tissue, which could manage an activation of the renewal of fatty acids with a specific rhythm, independently of the obesity status.”

- We know that adipose tissue composition is influenced by menopausal status. So how you discarded this parameter? 

This is a very good point. In this pilot study, we note 4 non-menopausal patients (2 aged 24 and 26 years and 2 aged 51 years). Removing these patients from the relapse analysis leads to similar results: low PUFA n6+n3 levels in subcutaneous adipose tissue (SAT, n=42) or omental adipose tissue (OAT, n=36) were associated to early relapses (see the graphs below).

Looking in detail at PUFAn-6, the contents are not different between non-menopausal and menopausal patients (mean 13.02 +/- 1.79 vs 12.96+/- 2.2), but 3non-menopausal patients have relapsed.

We propose to modify the discussion by adding this issue as a study limitation.

“Moreover, several studies (Bolton-Smith 1997), including the one we published on breast adipose tissue (Ouldamer 2022), show that PUFAs n-6 and n-3 levels slightly increase with age after menopause. These increases particularly concern long-chain PUFAs (> 20 carbons). In this pilot study, 4 patients were not postmenopausal and could confound the interpretation of the results. Relapse-free survival analyses performed by excluding these 4 patients, lead to identical conclusions with a significant decrease in RFS for patients with low PUFAn-6+3 levels (data not shown). However, to overcome this potential confounding factor of age, additional adipose fatty acid profiles will have to be acquired on non-menopausal patients.”

The idea of the study is very good but you need to broaden your discussion approach further and increasingly focus on the relationship between the fatty acid composition of adipose tissue and cancer.

All these remarks must be taken in consideration. Answers to these criticisms will reinforce the quality of the manuscript and will permit to conclude with more accuracy.

Round 2

Reviewer 1 Report

The paper was revised well.

Reviewer 2 Report

In this version of the article “Total polyunsaturated fatty acid level in abdominal adipose 1 tissue as an independent predictor of recurrence-free survival in 2 women with ovarian cancer”  We can see a great evolution compared to the first version, especially in the introduction and the discussion part, because they have become more structured, deeper and more scientific with more explanation.

the authors have considered the reviewer's remarks and suggestions, which has positively impacted the quality and consistency of the article.

with this version, the article shows a good scientific level and adds value to this research topic.

the article is accepted for me with this version